# A Systematic Review of Time Series Classification Techniques Used in Biomedical Applications

**DOI:** 10.3390/s22208016

**Published:** 2022-10-20

**Authors:** Will Ke Wang, Ina Chen, Leeor Hershkovich, Jiamu Yang, Ayush Shetty, Geetika Singh, Yihang Jiang, Aditya Kotla, Jason Zisheng Shang, Rushil Yerrabelli, Ali R. Roghanizad, Md Mobashir Hasan Shandhi, Jessilyn Dunn

**Affiliations:** Biomedical Engineering Department, Duke University, Durham, NC 27708, USA

**Keywords:** systematic review, time series classification, digital clinical measures, machine learning, feature engineering

## Abstract

*Background:* Digital clinical measures collected via various digital sensing technologies such as smartphones, smartwatches, wearables, and ingestible and implantable sensors are increasingly used by individuals and clinicians to capture the health outcomes or behavioral and physiological characteristics of individuals. Time series classification (TSC) is very commonly used for modeling digital clinical measures. While deep learning models for TSC are very common and powerful, there exist some fundamental challenges. This review presents the non-deep learning models that are commonly used for time series classification in biomedical applications that can achieve high performance. *Objective:* We performed a systematic review to characterize the techniques that are used in time series classification of digital clinical measures throughout all the stages of data processing and model building. *Methods:* We conducted a literature search on PubMed, as well as the Institute of Electrical and Electronics Engineers (IEEE), Web of Science, and SCOPUS databases using a range of search terms to retrieve peer-reviewed articles that report on the academic research about digital clinical measures from a five-year period between June 2016 and June 2021. We identified and categorized the research studies based on the types of classification algorithms and sensor input types. *Results:* We found 452 papers in total from four different databases: PubMed, IEEE, Web of Science Database, and SCOPUS. After removing duplicates and irrelevant papers, 135 articles remained for detailed review and data extraction. Among these, engineered features using time series methods that were subsequently fed into widely used machine learning classifiers were the most commonly used technique, and also most frequently achieved the best performance metrics (77 out of 135 articles). Statistical modeling (24 out of 135 articles) algorithms were the second most common and also the second-best classification technique. *Conclusions:* In this review paper, summaries of the time series classification models and interpretation methods for biomedical applications are summarized and categorized. While high time series classification performance has been achieved in digital clinical, physiological, or biomedical measures, no standard benchmark datasets, modeling methods, or reporting methodology exist. There is no single widely used method for time series model development or feature interpretation, however many different methods have proven successful.

## 1. Introduction

Time Series Classification (TSC) involves building predictive models that output a target variable or label from inputs of longitudinal or sequential observations across some time period [1]. These inputs could be from a single variable measured across time or multiple variables measured across time, where the measurements can be ordinal or numerical (discrete or continuous).

Time series data are a very common form of data, containing information about the (changing) state of any variable. Some common examples include stock market prices and temperature values across some period of time. Time series modeling tasks include classification, regression, and forecasting. There are unique challenges that come with modeling time series, given that measurements in time obtained in real-life settings are subject to random noise, and that any measurement at a particular point in time could be related to or influenced by measurements at other points in time [1]. Given this nature of time series data, it is impractical to simply utilize established machine learning algorithms such as logistic regression, support vector machine, or random forest on the raw time series datasets because these data violate the basic assumptions of those models. In recent years, two vastly different camps of time series classification techniques have emerged: deep-learning-based models vs non-deep-learning-based models. While deep learning models are extremely powerful and show great promise in classification performance and generalizability, they also present challenges in the areas of hyperparameter tuning, training, and model complexity decisions. To enable the evaluation of new models, a reasonable baseline is also needed for comparison. Further, there already exists a review on deep-learning time series classification methods [2]. Therefore, the focus of this review is on non-deep learning-based time series classification models.

This paper also focuses specifically on the biomedical applications of time series classification because there has been a huge increase in the generation of biomedical time series datasets (such as data from wearable devices like Apple Watch and Fitbit) recently as well as research using such data—examples include electrocardiogram (ECG, for cardiovascular dysfunction screening) [3], electroencephalogram (EEG, for brainwave tracking) [4], accelerometry (for activity recognition), and polysomnography (PSG for sleep tracking) [5], etc. In addition, an increasing number of people use smart devices or wearables regularly [6] for general fitness tracking [7], sleep tracking [8], fall detection [9], or arrhythmia detection [10]. There is a growing need to design better data mining and classification methods to discern important and useful information from biomedical time series data. This would lead to more reliable methods for screening, diagnosis, and monitoring, thereby providing huge benefits for healthcare as a whole.

Biomedical time series data collected from human subjects often present challenges that impede the ability to leverage time series modeling techniques that are common in other fields. For example, biomedical time series datasets often include just a small number of human subjects due to the resources and effort needed for data collection and annotation (or labeling to produce ground truth), which makes applying deep learning models very difficult since they are extremely data hungry [11]. Another challenge is the non-ergodic nature of datasets collected from human subjects, meaning that human subjects have vast individual differences in mental and physical states, and thereby producing data that look very different from one subject to another [12]. This results in sample level observations or models that perform well on some individuals even while being completely useless for others.

While both reviews and experimental evaluations of recent algorithmic advances have been done [13], the usefulness and applicability of machine learning algorithms is also impacted by interpretability and simplicity, particularly for biomedical predictive or diagnostic tasks. This review systematically surveys papers published in recent years that have used time series classification machine learning algorithms on biomedical datasets to answer the following questions:(1)What are the most common time series classification algorithms used in biomedical data science in the past six years?(2)What are the best performing time series classification algorithms for common biomedical signals?(3)How is interpretability addressed in the scientific literature that describes applying TSC algorithms for specific biomedical tasks?

The motivation for this review came from the observation that the types of algorithms explored and the depth of analysis performed in time series biomedical data science have not been well described. In general, there has been a strong emphasis on algorithmic performance and a lack of focus on interpretability and model simplicity. This review aims to provide a general and recent landscape of the types of time series classification algorithms on longitudinal biomedical data and these algorithms can be applied toward specific tasks and with more insightful analysis.

## 2. Methods

A list of search terms was developed that are specific for each of the following four databases: PubMed, IEEE, SCOPUS, and Web of Science (Appendix A). Four literature databases were searched: PubMed, IEEE, Web of Science, and SCOPUS. Among these databases, IEEE enforces a limit on the number of search terms to a maximum of 20. Hence, the defined search terms on IEEE were different from those on PubMed, Web of Science, and SCOPUS. The literature search was limited to the last six years for a manageable scope of review. The defined searches include the general terms and variations of time series machine learning classification and the fields of biomedical data. While the approach limits the coverage of the review, more recent work is often built upon previous work and new time series classification techniques are often compared to established techniques from previous work, therefore this method is expected to provide a sufficient representation of the field. Covidence was used for literature screening and data extraction. This review has a very clear focus on only non-deep-learning time series classification techniques utilized on biomedical data. This boundary notably excludes time series regression tasks and deep learning techniques.

There were two phases of screening before data extraction. The first phase was screening by titles and abstracts, which Covidence automatically extracted from the DOI URLs. This phase was completed by two reviewers where each reviewer read through each title/abstract and labeled them as “include” or “exclude”. Conflicts were resolved by discussion among the reviewers in order to reach consensus. A total of 260 papers were found to be irrelevant in this phase, mainly due to the following reasons:Classification algorithm is not used on biomedical data or time series data;The article does not focus on classification algorithms, but regression algorithms, clustering algorithms, or other algorithms;The article focuses only on deep learning algorithms.

The second phase was screening of the full texts, which were pulled automatically by Covidence if free full texts were readily available. The rest of the full texts were uploaded manually to the Covidence platform using university credentials for access. Again, two reviewers each went through all the papers and adjudicated the inclusion of papers into the final data extraction. Conflicts were resolved by discussion to reach a consensus. A total of 40 papers were excluded in this phase, mainly due to the following reasons:No access to full paper;Not enough information about classification algorithm performance included;The data came from animals instead of humans;The algorithms used are not classification algorithms.

Data from each paper were extracted by one of five reviewers, and then verified, edited, and cleaned by the study lead. In Table 1, we detail the information that was extracted from each paper:

## 3. Results

After removing duplicates and irrelevant papers, 135 articles remained for review and data extraction. Time series classification modeling typically consists of 3 main steps: signal preprocessing/transformation, modeling, and classification (Figure 1). The classification step is basically the process of model tuning, training, and validation. The different types of algorithms used in the modeling steps are adapted from the categories summarized in “The great multivariate time series classification bake off: a review and experimental evaluation of recent algorithmic advances” by Ruiz et al. [13] (Figure 1b). The most common techniques found in our search are feature engineering and selection, statistical modeling, distance-based, index development, and shape-based methods (Table 1 and Appendix A).

## 4. Algorithm Summaries

Among the articles reviewed, electroencephalogram (EEG) signals were the most common biosignals investigated. Detailed information about the types of signals investigated in these papers are shown in Figure 2a. Engineered time series features that are fed into widely used machine learning classifier models are the most commonly used technique and most often found achieve the best performance (77 out of 135 articles). Statistical modeling (24 out of 135 articles) algorithms are the second most common. Wavelet-based classification models (8 out of 135 articles) are also common (Figure 2c). Of papers that reported accuracy, 64% achieved accuracy higher than 90%. Of those that reported F1-scores, 70% achieved an F1-score higher than 0.90. Of those that reported AUC-ROC values, 24% achieved AUC-ROC values higher than 0.90. Of those that reported the sensitivity and/or specificity, 54% and 57%, respectively, achieved scores higher than 0.90. Of those that reported Cohen’s Kappa, 43% achieved a Cohen’s Kappa higher than 0.90.

The classification performance metrics of all the articles were recorded and included in this review, including accuracy, F1-score, Area Under Curve of Receiver Operating Characteristics (AUC-ROC), sensitivity, specificity, and Cohen’s Kappa. The accuracy score is the most commonly reported performance metric, with 68% of the articles reporting accuracy scores (Figure 3). All other performance metrics are seldom reported: 30% reported F1-score, 19% reported AUC-ROC, 35% reported specificity, 43% reported sensitivity, and 6% reported Cohen’s Kappa. This is concerning because oftentimes using only one or two performance metrics to evaluate a classifier is unreliable and does not tell the whole story of performance [21,22]. In the 135 papers reviewed, 86.7% reported one or more performance metrics, 50.4% reported two or more performance metrics, and 37.8% reported three or more performance metrics. Only 2 papers out of 135 reported more than 6 performance metrics.

### 4.1. Preprocessing Methods

In all 135 papers, 50% specifically mentioned the preprocessing methods used. The most common preprocessing method is filtering, which was used mainly for artifact removal or noise reduction. Some other common preprocessing methods include re-sampling (downsampling for lower frequency or upsampling for higher frequency), segmentation, and smoothing. Other common methods are the use of discrete wavelet transform to decompose the original signal into different frequency bands [23,24,25], the use of continuous wavelet transform to expand the feature space [26], and the use of Fourier transform for signal decomposition and feature extraction [27,28]. There are also intelligent upsampling techniques, such as the use of synthetic data generation for a larger sample during preprocessing [29]. We present a summary of the commonly used preprocessing methods in Appendix A.

### 4.2. Feature Engineering Methods

Feature engineering was the most commonly used method of time series classification. The feature engineering pipeline (Figure 1b) usually consists of the following steps:Preprocessing: this step takes raw data as the input and performs some manipulation of the data to return cleaner signals. Common steps include artifact removal, filtering, and segmentation.Signal transformation: this step can be used in preprocessing and also as a precursor to feature extraction. Some manipulation is performed on the signal to represent it in a different space. Common choices are Fourier Transform and wavelet transforms.Feature extraction: in this step, features are extracted from the time series data as a new representation of the original time series.Feature selection: this step selects the features that are the most descriptive, or have the most explanation power. Feature selection is also frequently performed in conjunction with model building.Model selection: the best model is found through hyperparameter tuning and/or comparisons between different types of algorithms.Model validation: performance metrics are calculated for all of the final models. This is frequently done in conjunction with model selection and often using some form of cross-validation.

An example feature engineering technique for a time series is shown in Figure 4. Summary tables of the extracted features for general time series data as well as specific signals (HRV, EEG, etc.) and feature selection methods are presented in Appendix A. Appendix A also presents a summary table for all of the found feature selection methods. In short, feature engineering is used for all signal types across many different applications.

### 4.3. Other Methods

Ensemble Methods: Ensemble-based methods are characterized by the connection of multiple algorithmic models that join forces to make the final prediction. These methods may or may not need an additional feature engineering step. Some algorithms that do not necessitate feature engineering in this category are Hierarchical Vote Collective of Transformation-based Ensembles and Bag of Symbolic Fourier Approximation Symbols ensemble algorithms (BOSS) [13]. Newman et al. [31] describe a novel 3-classifier ensemble algorithm for detecting short periods of artificially induced nystagmus (a vision condition in which the eyes make repetitive, uncontrolled movements) from continuous eye movement data. The ensemble of classifiers include a support vector machine (SVM), a linear discriminant analysis (LDA), and boosted trees, and the final classification decision is made by the majority vote. This method reported an accuracy of 0.9877, F1-score of 0.98, sensitivity of 0.9911, and specificity of 0.9863. Elsayed et al. [32] tested eight different state-of-the-art time series classification methods to find the optimal univariant ECG signal classifier. These models are: the Fully Convolutional Network (FCN), Long Short-Term Memory and Fully Convolutional Network (LSTM-FCN) and its attention-based LSTM model (ALSTM-FCN), the Deep Gated Recurrent and Convolutional Network Hybrid Model (GRU-FCN), the Residual Network Mode (ResNet), Multilayered Perceptron model (MLP), Dynamic Time Warping model (DTW), and the noise-reduction-based model, BOSS. The best performance resulted from GRU-FCN, which achieved the highest accuracy in five out of the six datasets that were tested, with a reported accuracy score of 0.92.

State-space Models: State-space models are characterized by the construction of a state and transition model where the transitions are modeled by probabilities. Often, state-space models are most intuitively used for sequence-to-sequence or point-wise classification. For example, She et al. [33] introduced an adaptive transfer learning algorithm to classify and segment events from non-stationary, multi-channel temporal data recorded by an Empatica E4 wristband, including 3-axis accelerometry (ACC), heart rate (HR), skin temperature (TEMP), and electrodermal activity (EDA). Using a multivariate Hidden Markov Model (HMM) and Fisher’s Linear Discriminant Analysis (FLDA), the algorithm adaptively adjusts to shifts in the distribution over time, thereby achieving an accuracy of 0.9981 and F1-score of 0.9987. Garcia et al. [34] proposed a method based on dynamic affect (or emotional state) recognition from multimodal physiological signals such as EEG, Electrooculography (EOG), and Electromyography (EMG). This model is based on learning about latent space using Gaussian Process Latent Variable Models (GP-LVM), which maps high-dimensional data (multimodal physiological signals) to a low-dimensional latent space. A support vector classifier is implemented to evaluate the relevance of the latent space features in the affect recognition process, thereby achieving an accuracy of 0.90556.

Shape/Pattern-based: These models are characterized by mining or comparing shapes or patterns in a time or sequence vector. For example, Zhou et al. [35] published an algorithm that can take into consideration the interaction among signals collected at spatiotemporally distinct points, where fuzzy temporal patterns are used to characterize and differentiate between different classes of multichannel EEG data. This algorithm achieved an accuracy of 0.9318 and an F1-score of 0.931, thereby classifying positive vs negative emotion states.

Distance-based: These models calculate the distance (or differences) of time series data vectors. For example, Forestier et al. [36] propose an efficient algorithm to find the optimal partial alignment (optimal subsequence matching) and a prediction system for multivariate signals using maximum a posteriori probability estimation and filtering. This scoring function is based on dynamic time warping. They were able to achieve an accuracy of 0.95, an F1-score of 0.926, and a sensitivity of 0.896.

Other: There are other methodologies that are difficult to characterize. One common method is performed by using statistical modeling of some sort. For example, İşcan et al. [37] published a high performance method to classify and discriminate various ECG patterns (to identify and classify QRS complexes). The model is called LLGMN, which is composed of a Log-Linear Model and a Gaussian Mixture Model (GMM), and gives a posterior probability for the training data. This model was able to achieve the highest accuracy, which was 0.9924.

Another common method is designing a composite metric or index based on domain knowledge or data-driven metrics. For example, Zhou et al. [38] proposed a new algorithm to detect gait events on three walking terrains in real-time based on an analysis of acceleration jerk signals with a time–frequency method to obtain gait parameters, as well as detecting the peaks of jerk signals using peak heuristics. The performance of the newly proposed algorithm was evaluated in eight healthy subjects walking on level ground, upstairs, and downstairs. The mean F1-score was above 0.98 for HS (heel-strike) event detection and 0.95 for TO (toe-off) event detection on the three terrains.

Some articles focus specifically on investigating the wavelet transform and increasing its usefulness for specific use cases. For example, Ji et al. [39] systematically investigated the performances of mother wavelets commonly used in detecting gait events. The overall performance of the Continuous Wavelet Transform (CWT) in detecting the two gait events was significantly different when using various mother wavelets. “Db6” has the highest detection accuracy with the lowest detection time-error, achieving a final accuracy of 1.0. Lu et al. [40] proposed two methods: Discrete Wavelet Transform (DWT) and Extra Trees Classifier, and a personal identification method based on Continuous Wavelet Transform (CWT) and Convolutional Neural Networks (CNN). Nested five-fold cross-validation was used for model selection and model assessment. The CWT method was adopted to uncover feature differences between EMG signals of different subjects. The two methods achieved accuracies of 0.99206 and 0.99203, respectively.

### 4.4. Interpretation Methods

Model interpretability is a significant aspect of model building. In time series classification for biomedical applications, the interpretation of models that have been built and validated could highlight potential insights into the biomedical phenomenon of interest. Some models have a built-in methodology of interpretation, such as statistical modeling (Hidden Markov Models, Bayesian Models, or ARIMA models) and indices that are informed based on domain knowledge. For many more models with great performance, however, interpretability is a challenge. Only 47 out of the 135 papers reviewed have included some form of interpretation method or model explanation method. Table 2 summarizes the different types of model interpretation methods with descriptions and some examples.

### 4.5. Best Performing Algorithms

While it is impossible to reasonably declare that one particular type of time series classification algorithm is best for all biomedical applications, it is possible to recommend certain algorithms that achieved great performances and are commonly cited for each different input data type. The algorithm(s) that achieved the best performances are selected and summarized for each of the following most common input signal types: Electrocardiogram (ECG), Electroencephalogram (EEG), Actigraphy, Electromyogram (EMG), Photoplethysmogram (PPG), and Inertial Measurement Unit (IMU). In these papers, wavelet transform processing combined with neural network classifiers achieved the best results for Actigraphy data. (Note: This neural-network-based model is included in the review because it is not exactly deep learning, given our focus on time series specific transformation techniques.) Overall, the statistical modeling classifiers and feature engineering methods performed the best and most consistently for all input signal types. We also observed that wavelet transformation was consistently used as a preprocessing method, feature extraction method, or as an integral part of index development, and furthermore, that it achieved great results.

Electrocardiogram (ECG): Adam et al. [70] introduced a time series classification algorithm that extracts 224 non-linear features and relative wavelet features, selects 15 features ranked by the ReliefF method, and uses the k-nearest neighbor as the classifier for the detection of cardiovascular diseases from electrocardiogram signals. This approach achieved an average accuracy of 0.9927, sensitivity of 0.9974, specificity of 0.9808, and positive predictive value (PPV) of 0.9952. This algorithm can be categorized as feature engineering combined with wavelet transform processing.

While the intention of this review is to find non-deep learning algorithms that can perform very well, some algorithms that cannot be completely classified as just deep learning models are also included, for example, when the neural network architecture is very small with a stronger focus on statistical modeling—or when it’s designed with a time series specific transformation. Here, two such algorithms are included for discussion that also perform very well for ECG signals. Iscan et al. [37] present a time series classifier model composed of a Log-Linear model and a Gaussian mixture model—short-handed as LLGMN. This is essentially a neural-network-based model that gives a posteriori probability for the training data. This algorithm was able to achieve a high accuracy of 0.9924. He et al. [71] presented an algorithm using Continuous Wavelet Transform and Convolutional Neural Networks to achieve the best accuracy, which was 0.9923, and an F1-score of 0.994, a sensitivity of 0.9941, and specificity of 0.9891.

Electroencephalogram (EEG): Newman et al. [31] described a novel 3-classifier ensemble algorithm for detecting short periods of artificially induced nystagmus from the long-term eye movement data collected by the CAVA, which achieved an accuracy of 0.9877, F1-score of 0.98, sensitivity of 0.99, and specificity of 0.9963. The frequency domain features were used and calculated using FFT. The ensemble of classifiers include an SVM, an LDA, and boosted trees, and the final classification decision is made by a majority vote. [An efficient automatic arousals detection algorithm in single channel EEG.] Ugur et al. [72] used a simple SVM classifier to detect arousal state. This algorithm was able to achieve an accuracy of 0.982, F1-score of 0.962, sensitivity of 0.9467, and specificity of 0.9933. The features that were extracted were the mean and the variance of the scalogram (CWT squared) coefficients for a range of 16–21 Hz.

Actigraphy: Casado et al. [73] examined various different methods to classify and recognize walking, which was captured by the inertial sensors (accelerometer, gyroscope, and magnetometer) of a mobile phone. The authors examined both feature-based techniques and shape-based techniques. For the shape-based techniques, the authors evaluated the subsequence dynamic time warping, support vectors of an SVM as representative patterns, Partitioning Around Medoids (PAM) as representative patterns, and supervised summarization. The shape-based techniques achieved the best accuracy, which was 0.9535, by using a support vector machine with an rbf kernel. Among the feature-based techniques, the best accuracy was achieved by a Random Forest model with an accuracy score of 0.9531. The Convolutional Neural Network models achieved the best accuracy of 0.9834, even with one input channel (as opposed to nine channels in the other models). Islam et al. [74] published an algorithm that achieved an accuracy of 0.9523, F1-score of 1.0, sensitivity of 0.75, and specificity of 0.8824 using a Random Forest algorithm. The 23 features used are the maximum, minimum, average, standard deviation, variance, coefficients of variations in duration (CVD) of stride, stance and swing intervals, age, approximate entropy (ApEn), weight, height, sex, and gait speed of participants.

Electromyograph (EMG): Lu et al. [40] presented an algorithm using EMG for personal recognition that uses feature engineering and the ExtraTrees Classifer, thereby achieving an accuracy of 0.99206. Meshab et al. [26] presented an algorithm for the prediction of recovery from spinal cord injury using EMG. This algorithm extracts features using the time-domain EMG total power and pattern variability, frequency-domain features computed using Fast Fourier Transform (FFT), Short-Time Fourier Transform (STFT), and Continuous Wavelet Transform (CWT), and makes predictions using kNN, thereby achieving an accuracy of 0.975.

Photoplethysmogram (PPG): She et al. [33] presented an algorithm that uses a multivariate Hidden Markov Model to adaptively learn the data distribution and a Linear Discriminant analysis to classify sleep vs wake from PPG data. This algorithm achieved an average accuracy score of 0.9981 and an F1-score of 0.9987.

Inertial Measurement Unit (IMU): Hemmati et al. [75] presented a wavelet-based algorithm to detect postural transitions. The inertial signal was decomposed using a 4^th^-order Daubechies Wavelet Transform and the classifier uses subject-specific fixed thresholds (curve length and area under the curve) to achieve an accuracy as high as 0.96. Pham et al. [68] presented an algorithm for the detection of steps using Continuous Wavelet Transform and found the minimum and maximum, thereby achieving an accuracy score of 0.99, sensitivity score of 0.9, specificity score of 0.88, PPV of 0.96, and NPV of 0.73. Martindale et al. [76] presented an algorithm for the prediction of activity levels using a hierarchical Hidden Markov Model, thereby achieving an F1-score of 0.962, a sensitivity score of 0.956, and a specificity score of 0.992.

## 5. Discussion

While deep learning methods have seen wide usage and high performance in health informatics in recent years, this review demonstrates the utility and power of non-deep learning machine learning algorithms. Many papers were reviewed with a focus on conventional machine learning algorithms that achieved almost perfect performance in classification metrics (i.e., 0.999 in classification accuracy). Compared to deep learning approaches, many conventional machine learning algorithms can be used off-the-shelf, without the researcher needing to rebuild the model architecture and tune a large number of hyperparameters. Conventional machine learning algorithms are also generally easier to train, optimize, and deploy due to their light-weight model (not necessarily needing a large number of parameters as in deep neural networks). This review also serves to identify the non-deep learning time series classification techniques that can serve as a competitive baseline comparator for researchers to understand whether newly designed deep learning networks are truly performing well or not. Among all of the papers reviewed, feature engineering methods followed by off-the-shelf machine learning techniques such as Support Vector Machine and Random Forest are by far the most common. To aid future researchers in building and testing feature engineering algorithms, we have provided an almost exhaustive list of features and transformation techniques that can be applied to longitudinal data in health informatics. A summary of the most common preprocessing methods has been provided, but we do not claim the summary to be exhaustive since preprocessing methods are very frequently domain dependent, and often decisions about preprocessing are made with the researchers’ own experiences and discretion with considerations about the different characteristics of each unique dataset. There is, however, a lack of standards in terms of the classification metrics reported, which goes against the best practices of reporting multiple metrics to fully describe the performances of the algorithms tested. A total of 42 out of the 135 papers that were reviewed only reported one metric, and 31 of these 42 papers reported the accuracy score, which is prone to bias [22,77].

### 5.1. Small Datasets

A pipeline of signal processing, feature engineering, and a classifier of choice were able to achieve high classification performances on datasets that came from small populations (<20). For example, Hong et al. [55] (as mentioned above) published an algorithm to detect drowsiness using EEG, PPG, and ECG signals. The data were collected from 16 healthy subjects, and non-linear features were extracted, selected, and fed into a Random Forest Classifier, thereby achieving an accuracy score of 0.99, F1-score of 0.99, and Cohen’s Kappa of 0.985. Among the papers that used small datasets, (discrete or continuous) wavelet transform—as a processing method or feature extraction technique—was very commonly used and very effective, such as for the paper presented by Hemmati et al. [75] (mentioned above), which used data from only 12 subjects. Statistical modeling methods are also very effective as classifiers, such as the paper presented by She et al. [33] (mentioned above), which used data from only 20 subjects. This highlights the benefits and present needs of non-deep learning time series classifiers, especially for biomedical applications where time series data with gold standard labels are difficult to come by.

### 5.2. Clinical Decision Support

Time series classification models are important for clinical decision support, being supplemented by Electronic Health Records (EHR) or other data that are gathered in the clinical setting to make predictions that could help healthcare providers better dedicate attention and resources, such as mortality rate predictions or early detection of sepsis. Again, a pipeline of preprocessing, feature engineering, and classifier models has been very effecitive, particularly because these kinds of models provide the ease of using domain knowledge in model building and have strong and intuitive interpretability. For example, Nancy et al. [56] presented a Statistical Tolerance Roughset-Induced Decision Tree (STRiD) using features that were extracted for the classification of subjects with hepatitis or thrombosis in a clinical setting—as opposed to without—thereby achieving an accuracy score of 0.915, F1-score of 0.9336, and AUC-ROC score of 0.93.

### 5.3. Medical Devices

Portable and wearable devices have developed stronger capabilities and gained wider usage over the years. Time series modeling using the continuous data stream that comes from these devices can generate medical insights over long stretches of time and identify digital biomarkers that can serve as a screening tool for common medical conditions [78]. Again, the feature engineering pipeline of time series classification is very commonly used and achieves great results due to the limited computational power and storage space found on these devices. An example of the application of time series classification modeling on medical devices is Newman et al.’s study [31], as presented above, which achieved an accuracy of 0.9877 and F1-score of 0.98.

## 6. Limitations

While rigorous, our paper selection method would have benefited from a third reviewer to break ties and resolve discrepancies. Furthermore, we were not aware of any existing classification system for categorizing the time series classification algorithms, and thus we developed our own, which may be sub-optimal. It is evident that many papers are difficult to categorize or assign a single category because studies often incorporate multiple different approaches, for example, using Dynamic Time Warping to calculate distances between time series motifs, and subsequently using those distances as input features into a Support Vector Machine. Additionally, although we sought to exclude deep learning approaches through our search term design, some papers examined both deep learning and non-deep learning classification algorithms and we felt compelled to include these papers in our review, both to not exclude the non-deep learning methods, as well as to gain insight into the direct comparison between these two approaches.

## 7. Conclusions

In conclusion, our group performed this systematic review to survey the landscape of non-deep-learning-based time series classification methods used in biomedical applications. Non-deep learning time series classification techniques can be extremely powerful—given their great algorithm performances—while also allowing for great interpretability. However, this field still lacks standardization for model testing and validation procedures and reporting metrics, which should be addressed to allow for better reproducibility and understanding of the algorithms that are presented by researchers in this field.

## Figures and Tables

**Figure 1 sensors-22-08016-f001:**
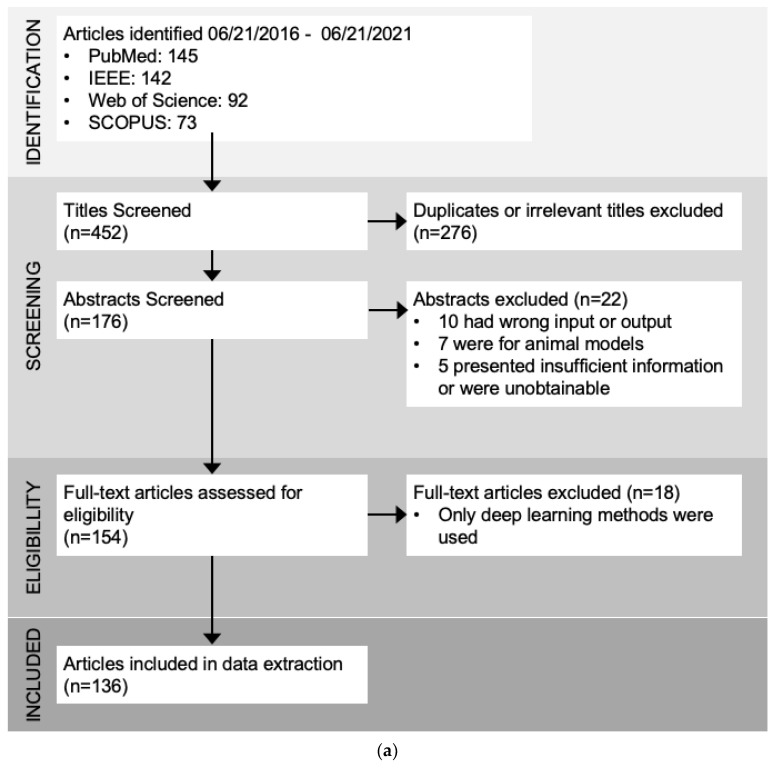
(**a**). Review results and the number of papers through each selection process. (**b**). Flow chart of the common steps in time series classification techniques found in this review. Raw time series signals usually go through some steps of preprocessing for artifact removal or noise reduction, and then are passed through the modeling stage. The modeling stage can use many different types of algorithms, such as feature engineering and selection, statistical modeling, and distance calculation (Table 1). Classifiers are then tuned, trained, validated, and compared to find the best model for a specific task.

**Figure 2 sensors-22-08016-f002:**
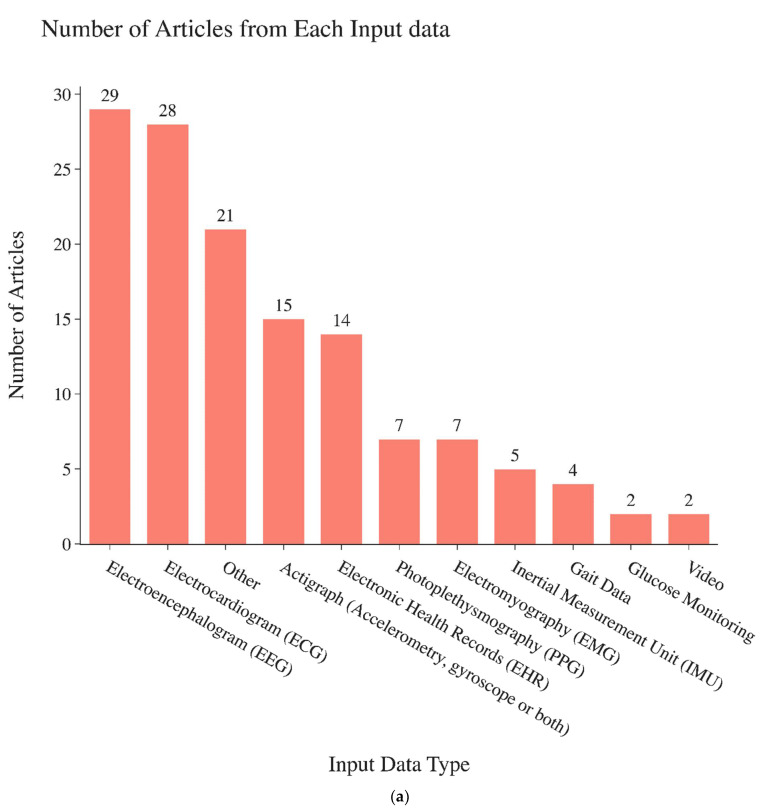
(**a**). Numbers of papers found in this review focusing on each different biosignal type specified on the horizontal axis. (**b**). Conceptual representation of non-deep learning time series classification modeling types. [1,16,17,18,19,20]. (**c**). Number of articles found for the categories of time series classification methods (horizontal axis) used in biomedical applications.

**Figure 3 sensors-22-08016-f003:**
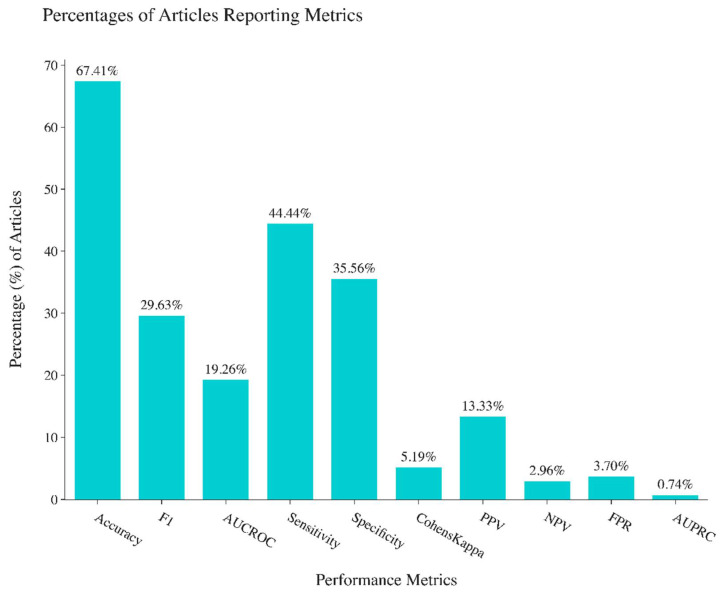
Percentages of performance metrics reported in studies reviewed.

**Figure 4 sensors-22-08016-f004:**
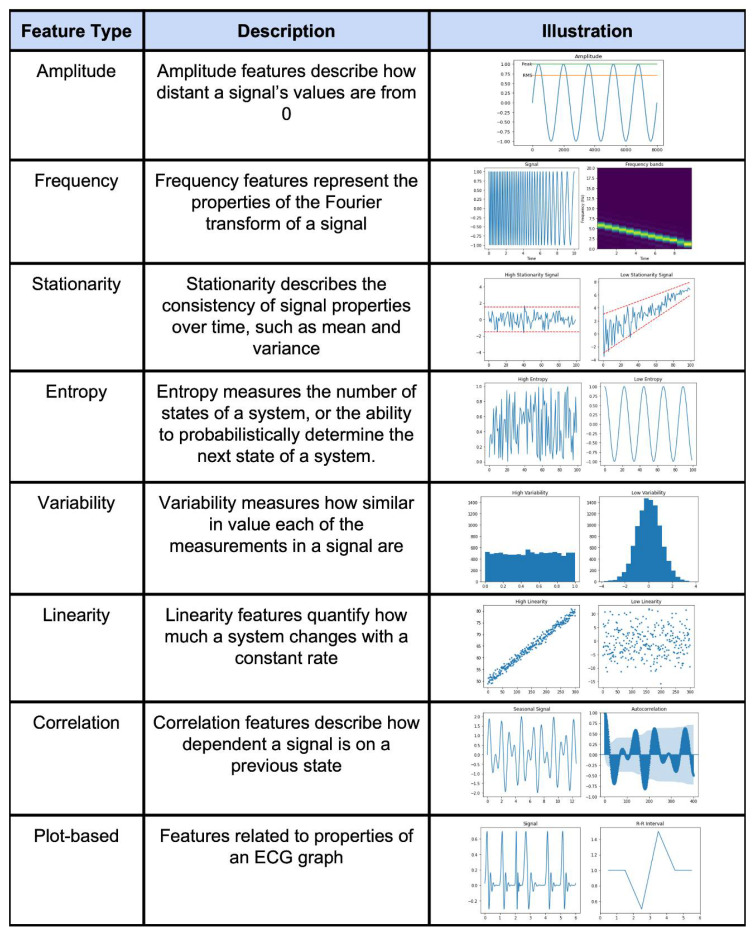
Illustration of different types of feature engineering techniques [30].

**Table 1 sensors-22-08016-t001:** Data fields extracted from identified academic research.

Categories	Choices/Sub-Fields	Definitions/Descriptions
General (Relevant) Information	Article Type	The type of article for a particular paper being reviewed, such as Journal Article/Conference Article/Review Paper
	Area of application	Describes the area of biomedical signal and application this paper is about.
	Aim of study	Defines the specific challenge or question this paper is aimed at tackling
	Name of Publisher/Journal/Conference	Site of article publication.
	Classification Task	Defines the kind of classification task performed in this article. (Pointwise classification, window classification, or whole sequence classification)
	Input data (X)	The type of input biomedical time series data
	Label (Y)	The output label or variable. Example: sleep vs wake, healthy vs diseased.
	Data source or open dataset name	States if the data are open source and where the dataset is hosted.
	Population Size	The number of subjects are included in this dataset.
	Data exclusion criteria	States the criteria considered to exclude subjects or specific parts of the data.
	All algorithms tested	List (or examples) of all the algorithms tested.
	Best algorithm name	The name of the best algorithm.
Classification Task	Whole-Series Classification	In whole time series classification (WSC) for a dataset of *n* samples, we are provided a set of tuples where each of an entire time series is associated with one class label.
	Sequence-to-sequence (point-wise)	The class label of each point in time is predicted.
	Window-based Classification or Onset Detection	Onset detection is a subtype of time series classification in which—as opposed to whole series classification—class labels are provided with a time-stamp. As an alternative to time pointwise classification, time-stamped labels have been leveraged for classifying time series windows that precede the class label’s time-stamp. For onset detection, a class label requires a time-stamp. This additional information can enforce that solely information from the past and present is used to predict a future target. This can be understood as a compromise between time pointwise classification and whole time series classification. An example is to detect the onset of sepsis in the intensive care unit
Best Algorithm Class	Feature Engineering	The type of time series classification technique where features are extracted to describe a particular time series sample and the features are fed into traditional machine learning algorithms as inputs of the predictive modeling.
	Statistical Modeling	This technique uses statistical modeling (such as Kalman filters or state-space models like Hidden Markov Models) to describe or fit the time series observed. Using the information obtained from statistical models, we can make decisions or extract features to be used as inputs to machine learning algorithms.
	Wavelet Transform [8]	Wavelet Transform can be used for signal cleaning (preprocessing), signal decomposition (preprocessing), and feature extraction. This technique is widely used and can be considered an integral part of time series machine learning.
	Distance-based methods [7]	This method is based on defining or quantifying the difference or distance (proxy for dissimilarity) between every pair of time series data samples in the dataset. Classification is performed based on the calculated distances, where two time series that are in close proximity (i.e., they have a small distance) under some distance measure are likely to come from the same class.
	Ensemble-based	Ensemble-based classification algorithms utilize multiple algorithms to make predictions and then aggregate the results coming from these different algorithms
	Shapelet/Shape-based	Shapelet-based methods are similar to significant pattern mining. Time series shapelets are subsequences that maximize classification performance.
	Non-linear index and thresholding	This time series classification method is based on defining indices based on domain- and data-driven time series features. The thresholds for these indices can be predefined or found through statistical learning. The thresholds are then used to make predictions of classes.
	Other	Any other methods of time series classification that cannot be easily categorized.
Best Algorithm Performances [14,15]	Accuracy	The degree of correctness of a calculation of the best algorithm reported. $\frac{TP+TN}{TP+FN+TN+FP}$
	F1-score	The harmonic mean of precision and recall of the best algorithm reported. $2\times \frac{Precision\times Sensitivity}{Precision+Sensitivity}$
	Area Under Curve of Receiver-Operating Characteristic	The measure of the usefulness of a test, in general, of the best algorithm reported.
	Sensitivity	The percentage of true positives of the best algorithm reported. $\frac{TP}{TP+FN}$
	Specificity	The percentage of true negatives of the best algorithm reported. $\frac{TN}{TN+FP}$
	Cohen’s Kappa	A statistical measure of inter-rater reliability for categorical variables of the best algorithm reported. $\frac{p_{0}-p_{e}}{1-p_{e}}$
	Positive Predictive Value	The percentage of positive test results is a true positive. $\frac{TP}{TP+FP}$
	Negative Predictive Value	The percentage of negative test results is a true negative. $\frac{TN}{TN+FN}$
	False Positive Rate	The percentage of false alarm of the best algorithm reported $\frac{FP}{FP+TN}$
	Area Under Precision-Recall Curve	A model performance metric for binary responses that is appropriate for rare events and not dependent on model specificity

**Table 2 sensors-22-08016-t002:** Summary of the different types of model interpretation methods discussed or used in each article.

Type of Interpretation Method	Description	Example Papers
Plotting and Annotating Raw Signal	Plotting and annotating raw signals is a widely adopted and useful method for explaining the significance of differentiating features or shapes in time series classification problems. The plots generally consist of a representation of the raw or preprocessed signals in scatter or line plots and highlight the characteristics of the raw or transformed signal, which serves as the differentiating features or shapes for different classes. Some groups have also adopted plotting of preprocessed and transformed signals to present interpretable results. Examples of this method include plotting heart rate values with steps that compare rest and active periods, plotting detected anomalous sequences that are compared against normal sequences, and plotting time series samples in cluster plots after dimension reduction or feature extraction.	[41,42,43,44,45,46]
Visualization of indices over biological/physiological constructs	Instead of plotting against raw signals in 1D, researchers also routinely plot calculated or estimated metrics against 2D or 3D biological constructs, especially when the time series data are signals that represent complicated biological systems, such as brain activities or blood circulation. This interpretation method is very commonly used on electroencephalogram datasets, and examples include a graphical representation of the brain for mean calculated metrics for calm and distressed individuals, as well as a construction of 2D maps of scalp topographies that indicate statistical differences.	[4,26,28,47]
Statistical Analysis/Modeling	Statistical analysis and modeling are used to provide interpretability for not just the models built for classification, but also for clinical application and biomedical understanding. Various plots and tests can be used to demonstrate the relationship between outcomes and certain features or estimated metrics. Example plots are kernel distribution plots, distribution box-plots from statistical models, normality plots, and the visualization of the separability of indices through plotting of the index space. Example analysis tests include variance analysis, normality tests, correlation analysis, and also modeling techniques such as generalized linear models, bivariate random-effects models, and Bayesian hierarchical models.	[26,41,48,49,50,51,52,53]
Feature weight/importance analysis/ranking	Analysis and visualization of feature importance in a model are very helpful for researchers and clinicians to identify the most useful and important features that contribute to predicting an outcome or influencing a diagnosis. Many time series classification algorithms have built-in methods for feature importance analysis, such as Random Forest, Logistic Regression, and some statistical modeling based classification algorithms. In the pipeline of feature engineering techniques of time series classification, it is often seen that feature selection or dimension reduction are used, and these steps also automatically generate a ranking of feature importance to model building. Additionally, feature importance and ranking can be generated by specific techniques such as Fisher Importance score and Shapley values.	[44,54,55,56,57,58,59,60]
Classifier Boundary Plotted against features	Plotting the classifier’s boundary in the feature space or lower dimensional space helps to visualize the classifier’s ability to differentiate observations from one class to another, i.e., separability. SVM-based classifiers commonly utilize this method for interpretability.	[61]
Index Parameter and Threshold Tuning	Index parameter and threshold tuning is an interpretation method that is usually used in conjunction with classifier building by using a domain-driven approach. Using a domain-driven approach, the researchers typically try to design an index to quantify a biological or physiological phenomenon. The design of the index is usually flexible and can be tuned by changing the parameters used in the index’s formula. The threshold of the index is used to differentiate the classes (such as normal vs abnormal conditions, positive vs negative diagnosis). Both the parameters and the threshold of the designed index can be tuned using the existing the dataset, and the classifier’s performance metrics can be examined to find the best set of parameters and threshold(s) to achieve the best classifier performances. These parameters and thresholds could also have biomedical significance and meaning relevant for future medical understanding and research. Index analysis can be performed against record length (length of time series), missingness, sample saturation, and time offset.	[51,57,62,63]
Channel or Signal Selection	Channel or signal selection is a model building technique but also an interpretation method. Given the prevalence of multivariable time series data in biomedical applications, it is critical for researchers to determine which signals among many, or which channels, can be used for classification. By comparing the classifiers’ performances using different signals/channels or specifications of the signals (such as where the sensors are placed), researchers are able to find the best combination that achieves the best performance results, and hence provides interpretability in terms of which signal types or channels are most important for the given biomedical application.	[64,65,66]
Performance Comparisons Investigating Different Scenarios	Comparing classifier performance metrics when built under different scenarios serves as an interpretation method as well as an experimentation method. Experts in a domain of interest can make sense of why a certain scenario produces the best predictability or algorithm performance, thereby contributing to biomedical understanding and research. For example, accuracy and F1-scores can be compared using datasets collected under different sensor inputs, different user locations, different symbolic or discretization methods, and different data fusion techniques	[60,67]
Bland–Altman plot illustrating the agreement	Bland–Altman plots can be used to evaluate the difference between estimated predictions from the algorithm and the gold standard, thereby providing interpretation of the algorithm’s prediction power and potential usefulness as a digital biomarker.	[68]
Deep Learning Network Analysis	Although deep learning models are generally thought of as black box models without easy and direct insight into what the models are doing, there has been recent and impactful research into developing the interpretability of deep learning models and some methods for model explanation. The cited example paper introduces a “global and local explanation”. Global explanation means looking at entire classes of data that show which regions of the signal patterns have the most influence for a specific class. Local explanation is the analysis of specific input signals and model outcomes. These methods enable a deeper understanding of the network’s behavior, thereby showing the most informative regions that trigger the classification decision and highlighting the possible causes of abnormal physiology or behavior.	[69]

## Data Availability

Not applicable.

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
