# Peer review of "A Systematic Review of Time Series Classification Techniques Used in Biomedical Applications"

_sensors, 2022, doi:10.3390/s22208016_

Round 1
Reviewer 1 Report
I apologize to the authors for being late - I tried but failed to get the review out before my holiday.
My concern is that how does one go from a laundry list of non-deep learning-based time series classification models applied to various signals to a story that captures the reader's attention. I have the following two suggestions: 1) your time window does not allow for measuring impact by quantifying the citations that reference these techniques. This may indicate the approaches have been most productive but on the other hand does anyone use techniques except the ones they developed? and 2) put yourself in the paper. Normally, I would not recommend this but what was your interest in doing this literature review? Do you plan to apply any of these approaches? If so, which ones? I liked your questions 1) What are the most common time series classification algorithms used in biomedical data science in the past six years? 2) What are the best performing time series classification algorithms for common 101 biomedical signals? 3) How is interpretability addressed ihe scientific literature that describes applying TSC algorithms for specific biomedical tasks?
but did not come away with answers for the last two.
How does your statement “Nearly half of the papers only report one metric, typically the accuracy score, which is prone to bias [15], [41].” fit with figure 3 (which I think should be reversed black for reporting and open for non-reporting)? Yes, accuracy score has the greatest white space but it is distinguish the overlap.
Author Response
- On your concern that we did not sufficiently address 2 questions laid out in the beginning of the paper, we added our answers in. In the results section, we added a summary of interpretation methods used in time series classification applied on biomedical applications (moved and edited from the supplementary materials), as well as a detailed subsection that points out the best algorithms used on the common signal types we found in the literature review. The best papers are selected based on a combination of citations, year published and performance metrics.
- On your question that Figure 4.b) did not prove what we wanted to show, that performance metrics of classification models are very often insufficiently reported, we recognized that the figure is not really helpful in making our point and deleted it from the manuscript. We instead explained this point with numbers and sentences.
Reviewer 2 Report
The manuscript entitled “A systematic review of tme series classification techniques used in biomedical applications” by Will Ke Wang et al. presents a comprehensive review of non-deep-learning time-series classification techniques that have been utilized for biomedical data for the past six years. It is well written, extensively and appropriately cites the literature, and within the scope of the journal. This manuscript would benefit from a few points discussed below:
1. Exemplifying scenarios in which non-deep learning machine learning algorithms are more powerful than deep-learning machine learning algorithms.
2. A specific discussion regarding clinical utilization of non-deep-learning machine learning algorithms-assisted biomedical devices for diagnosis and treatment decisions, as well as clinical translation challenges of these devices.
3. A table listing non-deep-learning machine learning algorithms-assisted biomedical devices that are currently being used for diagnosis and treatment decisions in the clinic.
Author Response
- We included examples and a short discussion when non-deep learning models are able to perform extremely well for very small datasets.
- We included examples of time series classification/analysis methods used for clinical decision support that achieved great performance in the discussion section, pointing out trends of what are commonly used effective methods.
- We included examples of time series classification methods used for biomedical devices that achieved great performance in the discussion section, also pointing out trends of commonly used effective methods.
Reviewer 3 Report
The topic is interesting and need of modern era. I have some suggestions to make it better.
1. Please precise the abstract and also mention the problem statement. The results in abstract comprises of 13 lines, please precise it in 2 or 3 lines. Also please make conclusion in one precise line. It will give the better picture. Instead of using "we", present in some other way, please rephrase it accordingly.
2. In line 54, please don't use term "we" in introduction section. Please rephrase it.
3. In section, materials and methods, also please rephrase first paragraph. Used of "we" is not a scientific way to represent.
4. Please elaborate conclusion section and add some future recommendations.
Best regards
Author Response
- We shortened Abstract and Conclusion to the best of our ability and substituted usage of “we” to the best of our ability.
- Rephrased.
- Rephrased.
- More recommendations and discussions are added, according to advice from other reviewers 1 and 2.